# Fresh-Cut Eruca Sativa Treated with Plasma Activated Water (PAW): Evaluation of Antioxidant Capacity, Polyphenolic Profile and Redox Status in Caco2 Cells

**DOI:** 10.3390/nu14245337

**Published:** 2022-12-15

**Authors:** Ileana Ramazzina, Veronica Lolli, Karen Lacey, Silvia Tappi, Pietro Rocculi, Massimiliano Rinaldi

**Affiliations:** 1Department of Medicine and Surgery, University of Parma, Via Gramsci 14, 43126 Parma, Italy; 2Department of Food and Drug, University of Parma, Parco Area delle Scienze 27/A, 43124 Parma, Italy; 3Department of Agricultural and Food Sciences, Alma Mater Studiorum, University of Bologna, Piazza Goidanich 60, 47521 Cesena, Italy

**Keywords:** PAW, rocket salad, antioxidant capacity, flavonoid glycosides, glucosinolates, redox status

## Abstract

Plasma Activated Water (PAW) has recently emerged as a promising non-chemical and non-thermal technology for the microbial decontamination of food. However, its use as a replacement for conventional disinfection solutions needs further investigation, as the impact of reactive species generated by PAW on nutritional food quality, toxicology, and safety is still unclear. The purpose of this study is to investigate how treatment with PAW affects the health-promoting properties of fresh-cut rocket salad (*Eruca sativa*). Therefore, the polyphenolic profile and antioxidant activity were evaluated by a combination of UHPLC-MS/MS and *in vitro* assays. Moreover, the effects of polyphenolic extracts on cell viability and oxidative status in Caco2 cells were assessed. PAW caused a slight reduction in the radical scavenging activity of the amphiphilic fraction over time but produced a positive effect on the total phenolic content, of about 70% in PAW-20, and an increase in the relative percentage (about 44–50%) of glucosinolate. Interestingly, the PAW polyphenol extract did not cause any cytotoxic effect and caused a lower imbalance in the redox status compared to an untreated sample. The obtained results support the use of PAW technology for fresh-cut vegetables to preserve their nutritional properties.

## 1. Introduction

Rocket, also known as arugula, refers to a group of plant species distinguished by their pungent-tasting leaves. *Eruca sativa* L. is the species most commonly used for human consumption (rocket salad). The main phytochemicals found in the different parts of rocket tissue that contribute to its antioxidant properties are phenolic compounds and flavonoids. Glucosinolates, which are sulfur-containing plant secondary metabolites, are responsible for the bitter taste of rocket salad and have shown antibacterial, anticarcinogenic, and antioxidant properties [1]. 

Many phenolic compounds are antioxidants that may aid in the prevention of human diseases (e.g., cancer and heart diseases), and they also possess immunomodulatory activity. The health-promoting effects of a vegetable-rich diet have partly been attributed to an increased intake of phenolic compounds with a high antioxidant capacity [2,3].

The processing of fresh leafy vegetables (e.g., salad) for the preparation of fresh-cut products has different effects on the antioxidant properties of the tissue, depending on the species used. On an industrial scale, it is common practice to sanitize fresh-cut vegetables with a chemical disinfectant (usually sodium hypochlorite) to remove pathogenic and spoilage microorganisms. However, concerns about human health and environmental pollution have led to the search for alternatives to chemical treatments that preserve the nutrient density of the materials [4]. 

As the demand for fresher, safer, and nutritionally dense foods has increased, non-thermal treatment technologies, such as hydrostatic pressure, pulsed electric fields, ultrasound, and plasma, have been considered [5]. Plasma is a partially ionized gas composed of electrons, ions, uncharged neutral particles (e.g., atoms, molecules, radicals), and ultraviolet photons [6]. 

Plasma Activated Water (PAW) is widely regarded as a promising method for the microbial disinfection of food [6]. To produce PAW, the water is subjected to a cold plasma discharge above or below the water’s surface. The reaction species generated by the plasma interact with the water molecules to initiate a variety of chemical reactions, resulting in a one-of-a-kind mixture of biochemically reactive chemicals. In the absence of other chemicals, a unique transfer of energy and chemical reactivity occurs from the gaseous plasmas to the water, leading to a product characterized by a remarkable, transient, broad-spectrum biological activity [7,8]. Therefore, PAW is a sustainable potential strategy for a variety of biotechnological applications, including water purification and biomedicine.

Studies have been conducted to evaluate the effectiveness of cold plasmas or PAW technology in inactivating microorganisms [9,10,11,12]. PAW has been shown to be effective in inactivating both natural microbiota and intentionally contaminated pathogens; however, in addition to the microbial quality, the effects of the treatment on other parameters need to be carefully evaluated.

The bioactive components of fresh fruits and vegetables have a significant impact on human health, mainly because they possess antioxidant abilities. As a result, for product quality, the preservation of these components is critical.

After washing with PAW, a significant increase in antioxidant activity was observed in fresh-cut apples [13], pears [14], and mushrooms [15]. Previous research has shown that exposure to reactive oxygen and nitrogen species (RONS) generated by cold atmospheric plasma treatment (CAP) can lead to the oxidation of some phenolic compounds in leafy vegetables [16]. An increase in some specific phenolic compounds after PAW treatment has been observed in different products such as blueberries [17], apples [18], and mung bean sprouts [19]. The authors suggested that this effect was due to a physiological response of the tissue to the stress caused by the reactive species. However, an increase in exposure time triggered oxidative reactions and a progressive reduction in the phenolic content and antioxidant activity.

In a recent study [16], we investigated the effects of PAW, generated by a high-power atmospheric pressure corona discharge plasma source, on the microbial flora of arugula. We found that PAW was able to decontaminate this product while causing only minor changes in the quality parameters. PAW treatments were found to be more effective against the targeted background microbiota compared to hypochlorite, selected as the reference sanitizer due to its widespread use in the food industry. Specifically, shorter immersion times were required to significantly reduce the populations of Enterobacteriaceae and psychotropic bacteria, and all the groups of spoilage microorganisms were inactivated after 2 min of arugula dipping in PAW. 

Considering these previous results, the aim of the present study was to investigate the effect of the same PAW treatments on some of the antioxidant properties of rocket salad. Specifically, in rocket salad washed in PAW, we explored (i) the antioxidant activity measured with an *in vitro* multimodal approach; (ii) the quali-quantitative content of polyphenolic compounds using the standard UHPLC-MS/MS technique; and (iii) the role exerted by polyphenolic extracts on cell viability and oxidative status in Caco2 cells by comparison to the untreated sample.

## 2. Materials and Methods

### 2.1. Chemicals

Analytical-grade chemicals were purchased from Sigma-Aldrich (Steinheim, Germany), except for methanol and hydrochloric acid, purchased from Romil (Feltham, UK).

### 2.2. Raw Material, Handling, and Storage

Rocket leaves were obtained from a fresh market (Cesena, Italy) and kept under refrigeration (2 ± 1 °C) for about 24 h before the experiment. Homogeneous, defect-free leaves were selected for the treatment and divided into 10 sub-samples.

### 2.3. Plasma Activated Water (PAW) Generation, Sample Preparation, and Plasma Treatments

PAW was obtained using distilled water through a corona discharge plasma source (prototype from AlmaPlasma s.rl., Bologna, Italy) as previously described [16]. A microsecond pulse generator (AlmaPulse, AlmaPlasma s.r.l.) was connected to a stainless-steel pin-electrode held 5 mm from the surface of 450 mL of distilled water and continuously stirred. The operating parameters were 9 kV peak voltage and 5 kHz frequency. The PAW was generated by exposing the distilled water to plasma for 4 min. The measured concentrations of H_2_O_2_, NO_2_^−^, and dissolved O_3_ after 4 min were 4.5 ± 0.1 mg/L, 30.4 ± 0.9 mg/L, and 0.3 ± 0.1 mg/L, respectively, and the pH of the PAW was 3.3, as reported by [16]. Immediately after the PAW generation, the rocket samples were immersed for 2, 5, 10, and 20 min in the PAW at room temperature. For each treatment, 20 g of the samples were immersed in 400 mL of PAW (product:liquid ratio of 1:20 (w:v)) and kept under constant agitation. Each treatment was repeated twice for two independent replications. To assess the effect of the PAW washing, immersion in sodium hypochlorite (NaClO) 100 ppm for 2 min was used as the standard reference washing procedure (CL). Moreover, the untreated samples (UT) were considered as controls. After immersion, samples of the two washing replicates were dried with adsorbent paper and immediately freeze-dried. The dried samples were stored at −20 °C until analysis.

### 2.4. Total Phenolic Content and Antioxidant Activity

The total phenolic content (TPC) and antioxidant activity of the rocket salad samples were determined by different microplate assays. The TPC was quantified using the Folin–Ciocalteu phenol reagent. The antioxidant activity was assayed with ABTS (2,2′-azinobis-3-ethylbenzothiazoline-6-sulfonic acid), DPPH (2,2-diphenyl-1-picrylhydrazyl) and FRAP (ferric reducing antioxidant power). All these experimental protocols were performed as previously described [20]. 

### 2.5. Polyphenols Extract Preparation for UHPLC-MS/MS Analysis and Cell Line Experiments

We chose to focus our analysis on the PAW-20 extract because the data from antioxidant activity assays have shown that this washing time resulted in an increase in the TPC of the amphiphilic fraction. Moreover, the previous data reported that this treatment time significantly reduced the microbial load and significantly increased the total flavonoid content of *E. sativa* and its extract, respectively [16].

Three g of treated (PAW) or untreated (UT) freeze-dried rocket leaves powder were mixed with 20 mL of 60% methanol, and the suspension was vortexed vigorously for 2 min. The sample was centrifuged at 10,000 × g for 10 min at 10 °C; then, the supernatant was collected while the pellet was extracted a second time. The supernatants of the two extractions were combined and the solvent was removed using a rotary evaporator (mod. Laborota 4001, Heidolph Instruments, Schwabach, Germany) at 35 °C. A cell-cultured medium containing 0.5% DMSO (pH 7.1) was used to dissolve the dry residue, which was stored at −80 °C (stock solution containing 500 mg of freeze-dried rocket leaves powder/mL) for further analysis. Two independent extractions were performed for each sample.

### 2.6. UHPLC-ESI-MS/MS Analysis

An ultra-high-performance liquid chromatography (UHPLC) system combined with a negative electrospray ionization (H-ESI II) triple-quadrupole mass spectrometer (Thermo Scientific, TSQ Vantage, Waltham, MA, USA) was employed for the quali-quantitative determination of phenolic compounds in the rocket salad extract. For these experiments, a SUNSHELL C18 (2.1 i.d. × 100 mm) column with a 2.6 μm particle size (Chromanik, Osaka, Japan) was used.

The sample (500 mg of lyophilized rocket leave powder/mL) was diluted with acidified water (0.2% formic acid) to achieve a final concentration of 3 mg/mL. The mobile phase (flow rate at 0.35 mL/min) consisted of water + 0.2% formic acid (eluent A) and acetonitrile + 0.2 formic acid (eluent B). For gradient elution, a 9-min linear gradient of 2 to 20% acetonitrile in 0.2% aqueous formic acid was used. The capillary temperature was set at 270 °C; the sheath and auxiliary gases were 40 and 5 arbitrary units, respectively; and the voltage source was 3 kV. For the MS/MS analysis, a vaporizer temperature of 200 °C argon was used, with a collision pressure of 1.0. 

For compound identification, a full-scan analysis with a range from m/z 100 to 1500 was employed, while a product ion scan experiment was performed for the not fully identified ions by using the full-scan method. Then, the mass spectra were compared with the literature data [21] and MS spectral databases [22,23]. The method of calibration curve was adopted for quantification of the flavonol glycosides by using rutin (external standard) calibration solutions at five concentration levels with a range of 0.1–10 µg/mL.

### 2.7. Cell Culture and Treatments

Caco2 cells were purchased from ATCC and grown in a 1:1 mixture of Ham’s F12:DMEM medium, supplemented with 10% fetal bovine serum (Lonza, Basel, Switzerland), 2 mM L-glutamine, 100 U/mL penicillin, and 100 μg/mL streptomycin, at 37 °C under a 5% CO_2_ atmosphere. A trypsin/EDTA (Sigma-Aldrich, Steinheim, Germany) treatment was used for cell harvesting. For the reactive oxygen species (ROS) and nitric oxide (NO) determinations, the Caco2 cells were grown in a 1:1 mixture of Ham’s F12:DMEM medium without red phenol (Sigma-Aldrich, Steinheim, Germany). The polyphenol extracts from the PAW, CL, and UT rocket leaves were diluted in a complete cell medium to the final concentration required for each experiment (0.1% maximum concentration of DMSO). The concentrations were referred to the extract stock solution that contains 500 mg of freeze-dried rocket leaves powder/mL. A medium containing 0.1% DMSO was used for the culture of the control cells.

#### 2.7.1. Assessment of Cell Viability

Caco2 cells were seeded at a density of 4 × 10^4^ cells/well in a white, clear-bottomed 96-well microplate and allowed to attach overnight. Increasing concentrations of polyphenol extracts (corresponding to 0.01–100 mg of freeze-dried rocket leaf powder/mL) from rocket leaves exposed to PAW washing for 20 min (PAW-20) or to the UT sample were used to treat the cells. After 5 h of incubation, cell viability was determined with the CellTiter-Glo^®^ Luminescent Cell Viability Assay (Promega, Madison, WI, USA), in accordance with the manufacturer’s protocol. The luminescence intensity was assessed with an EnSpire^®^ multimode plate reader (PerkinElmer, Waltham, MA, USA). The samples were derived from two independent extraction procedures. In each experiment, each sample was analyzed in quadruplicate and the data are reported as the mean ± standard deviation (SD) of the two independent experiments. 

#### 2.7.2. Assessment of Reactive Oxygen Species (ROS)

Caco2 cells were seeded at a density of 4 × 10^4^ cells/well in a black, 96-well clear-bottomed microplate and allowed to adhere overnight. ROS production was assessed using the DCFDA Cellular ROS Detection Assay Kit (Abcam, Cambridge, UK), following the manufacturer’s protocol. Intracellular esterases deacetylate the 2,7-dichlorofluorescein diacetate (DCFDA) into a non-fluorescent compound, which is subsequently oxidized into the fluorogenic dye 2′, 7′-dichlorofluorescein (DCF) by reactive oxygen species. Briefly, the cells were loaded with 20 μM DCFDA for 45 min at 37 °C and washed twice with PBS. Then, the Caco2 cells were treated with increasing concentrations of polyphenol extracts from the PAW-20 or UT samples for 5 h. The fluorescence intensity of the DCF (excitation 485 nm; emission 535 nm) was assessed with the EnSpireTM multimode plate reader (PerkinElmer, Waltham, MA, USA). Tert-Butyl Hydrogen Peroxide (TBHP), at a concentration of 100 or 150 µM, was used as the positive control for the experiments. The data were reported as a percentage of the control after the subtraction of the background (blank wells with no cells and compounds at the same concentration used for treatment), followed by normalization to total protein content quantified by the Bio-Rad DC Protein assay (Bio-Rad Laboratories, Hercules, CA, USA Bio-Rad). The samples were derived from two independent extraction procedures. In each experiment, each sample was analyzed in quadruplicate and the data are reported as the mean ± standard deviation (SD) of the two independent experiments.

#### 2.7.3. Assessment of NO

Caco2 cells were seeded at a density of 6 × 10^5^ cells/well in 6-well plates and allowed to adhere overnight. Increasing concentrations of polyphenol extracts from the PAW-20 or UT rocket leaves were used to treat the cells. After 5 h of incubation, the total intracellular nitrite/nitrate concentration was assessed using the Nitric Oxide Assay Kit (Abcam, Cambridge, UK), in agreement with the manufacturer’s protocol. Because NO is rapidly converted to nitrite and nitrate, their total concentration is used as a measure of NO production. The experiments were performed using 30 μL of cell lysate and the samples were incubated for 4 h with the enzyme nitrate reductase to allow the conversion of nitrate to nitrite. The fluorescence intensity of the DAN probe (excitation 360 nm; emission 450 nm) was assessed with the EnSpireTM multimode plate reader (PerkinElmer, Waltham, MA, USA). The samples were derived from two independent extraction procedures. In each experiment, each sample was analyzed in quadruplicate and the data are reported as the mean ± standard deviation (SD) of the two independent experiments. 

### 2.8. Statistical Analysis

SPSS statistical software (version 21.0, SPSS, Inc., Chicago, IL, USA) was used to perform the statistical analyses. The data obtained from the *in vitro* experiments were analyzed by the one-way Analysis of Variance (ANOVA) to evaluate the effect of the treatments on the measured variables, and the two-tailed Student’s t-test and/or Tukey’s HSD post hoc test were carried out for comparing the groups of interest. The data obtained from the cell line experiments were analyzed by pairwise multiple comparisons for one-way ANOVA, followed by Tukey’s HSD post hoc test to detect differences between the groups of interest (treatment vs. control and PAW-20 vs. UT at different concentrations). Statistical significance was determined with a conventional *p* value of ≤ 0.05.

## 3. Results and Discussion

### 3.1. Antioxidant Activity of Rocket Salad upon Exposure to PAW

Phenolic compounds are among the most important phytochemicals that possess antioxidant activity due to their chemical structure, which gives them redox properties and radical scavenging activity [24]. Therefore, in this study, we first investigated the antioxidant activity in rocket salad upon exposure to PAW for different times (2, 5, 10, and 20 min) or to washing with a hypochlorite solution (CL), the latter being a common reference method in the food industry. We also analyzed untreated rocket salad (UT) as a control sample. Antioxidant activity was investigated by a multimodal *in vitro* approach, according to our previous studies [20,25], to evaluate both the radical scavenging activity (RSA), i.e., DPPH and ABTS assays, and reducing power, i.e., total phenolic content (TPC) and FRAP. The results are reported in Table 1.

The ABTS assay (expressed as Trolox equivalent, TE) was performed on the hydrophilic and amphiphilic extracts, both of which showed comparable RSA for the UT and CL samples, whereas significant differences appeared, especially between the PAW washing times. Specifically, the RSA of the hydrophilic fraction was significantly higher (*p* ≤ 0.05) after the 10- and 20-min immersion of the arugula in the PAW than after the 2-min immersion, reaching values that were about 30–40% lower than those of the controls (UT and CL, respectively). In contrast to the TPC results (see below), increasing the treatment time seemed to decrease the RSA of the amphiphilic fraction. In fact, the RSA showed the highest value after dipping rocket salad for 2 min in PAW (65 ± 3 µmol TE g^-1^ DM), resulting in a mean increase of 40% compared to the controls (UT and CL), while it significantly decreased by 40% after 20 min compared to the 2 min treatment (*p* ≤ 0.05). 

These results on the RSA of the amphiphilic extracts were also observed for the DPPH assay, which evidenced a significant increase (*p* ≤ 0.05) in RSA after dipping the rocket salad in PAW for 2 min compared to the other treatment times (especially at 20 min) and the controls (both CL and UT).

Then, the evaluation of TPC in the rocket salad was performed by measuring the ability of both the hydrophilic and amphiphilic fractions to reduce the Folin–Ciocalteu reagent. The results showed that the shorter PAW washing time (2 min) significantly decreased the TPC in the hydrophilic extract (*p* ≤ 0.05), while the extension of the exposure time (from 5 to 20 min) did not significantly affect the TPC compared to the controls (both UT and CL). Regarding the amphiphilic fraction, which generally showed a higher reducing power than the hydrophilic fraction, the highest TPC value was observed for the CL sample. The PAW resulted in a significant increase in TPC value of about 70% after washing for 20 min compared to the UT sample. A previous study reported that PAW washing did not result in significant differences in the TPC compared to the untreated sample, except after 5 min of treatment, which induced a slight reduction [16]. It is noteworthy that the authors evaluated the TPC on the total ethanol/formic acid extracts, whereas, in this study, we analyzed both the hydrophilic and amphiphilic fractions separately. The TPC and FRAP assays performed with the amphiphilic fraction were in agreement, indicating a positive relationship between the reducing power and the exposure time to the PAW. 

Overall, antioxidant activity in terms of RSA was positively affected by dipping the samples for a short time in PAW, while increasing of the exposure time (20 min) seemed to increase the reducing power, due to an increase in TPC compared to the UT sample. This effect has also been previously observed by other authors and several explanations have been put forward. On the one hand, an increase in polyphenols can be attributed to the activation of key enzymes involved in the phenolic pathway after a PAW long-time exposure, as reported for fresh-cut rocket as a response to processing stress (e.g., cutting) [26]. On the other hand, the cell wall modifications caused by ozone treatment are believed to be responsible for the release of the conjugated phenolic compounds in the cell wall of fruits such as bananas and pineapples [27]; therefore, this might also happen in the case of PAW since ozone is one of its reactive species. It is generally believed, however, that the changes are related not only to the total amount but also to the type of phenolic compounds [28].

### 3.2. Qualitative and Quantitative Analysis of E. sativa Extracts

To increase the knowledge about PAW technology in the food matrix, we evaluated its effect on the qualitative and quantitative polyphenolic profile in the rocket salad samples. To this aim, a UHPLC-MS/MS analysis was performed on the methanolic extracts obtained after exposing the rocket salad to PAW for 20 min (PAW-20) or the UT sample.

The base peak chromatograms acquired in the full-scan mode of the analyzed samples are shown in Figure 1, and the MS data for the identified compounds are listed in Table 2. The chromatographic peak detection and sample data analysis are reported in more detail in Appendix A. 

Most of the compounds detected in both polyphenol extracts correspond to glycosylated flavonols, especially kaempferol, isorhamnetin, and quercetin, in agreement with the previous literature data [21]. As shown in Figure 1, quercetin-O-trihexoside (peak 8, m/z 787) and quercetin-O-dihexoside-O-sinapoyl-O-hexoside (peak 11, m/z 993) are the most abundant compounds in terms of relative abundance (about 20% each) in both samples. Next, the absolute concentrations of flavonol glycosides in the samples were calculated using the linear regression equation (y = 72,311x + 9283.7, R^2^ = 0.9985) obtained from the analysis of the rutin calibration solutions described previously. 

The results, reported in Table 3, showed similar profiles in terms of concentrations (µmol/L) of the total flavonol glycosides in both the analyzed samples, and differences were found in only two compounds, namely isorhamnetin-O-hexoside and quercetin-O-dihexoside, which were significantly higher in the UT sample than in the PAW-20 extract (*p* ≤ 0.05).

In addition, in these analytical conditions, glucosinolates were detected, such as glucoraphanin (1) and glucoerucin (2), which are mainly found in cruciferous vegetables as secondary metabolites and are responsible for the pungent aroma of rocket salad [30]. For comparison, their relative abundances were obtained as a signal ratio to the total chromatographic area, and it was found that both [M-H]^−^ glucoraphanin (1) and [M-H]^−^ glucoerucin (2) were significantly higher (*p* ≤ 0.05) in the PAW-20 (26 ± 2 and 24 ± 3%, respectively) than the UT sample (17.9 ± 0.8 and 15 ± 1%, respectively), corresponding to increases of about 44 and 50%, respectively. Interestingly, glucosinolates are the precursors of bioactive compounds, such as sulforaphene and erucin, which are extensively studied for their health-promoting effects [31,32,33]. Changes in the hydrolysis products of glucosinolates, such as isothiocyanates and nitriles, in rocket have been previously observed as a consequence of PAW treatment [34]. These compounds are obtained by the enzymatic hydrolysis of the respective glucosinolate compound. The authors observed a modification of the relative amounts of the hydrolysis products; that, however, was not time dependent. 

The increase in glucosinolate compounds could be related to the physiological response of the tissue to abiotic stress represented by washing in PAW. Other authors [35] observed an increase in the endogenous production of these metabolites in response to environmental stress. Indeed, abiotic stress can induce specific responses at a cellular level, which have the aim of counteracting the stressful conditions. Although the mechanisms have not been fully elucidated, often these responses can involve the de novo synthesis of secondary metabolites such as glucosinolates, and we can therefore speculate that a similar effect occurred in the present research. Moreover, for a better understanding of the health properties of the treated products, it would be interesting to investigate how PAW affects the further enzymatic hydrolysis of glucosinolates into their corresponding products. 

So, the results obtained by the UHPLC-MS/MS analysis showed only minor differences between the polyphenolic profile of the PAW-20 and UT samples, while washing in PAW significantly affected the spectrophotometric determination of TPC. This could be because the TPC assay depends on an oxidation/reduction reaction. Moreover, the two determinations were performed on *E. sativa* extracts obtained according to two different protocols of extraction, as specifically reported in the Materials and Methods section. Of note, the extracts characterized by the UHPLC-MS/MS analysis were used for the assays performed in the Caco2 cell line. 

### 3.3. Effect of PAW-Rocket Salad Extract on Cell Viability

We have previously reported that polyphenol extract from apples’ exposure to atmospheric double-barrier discharge (DBD) plasma technology did not affect cell viability [25]. To evaluate whether washing in PAW induces the generation of compounds that may be dangerous to human cells, first we evaluated whether the PAW-20 extract affects cell viability by using the CellTiter-Glo^®^ Luminescent Cell Viability Assay (Promega, Madison, USA). Caco2 cells were treated for 5 h with different concentrations of the PAW-20 or UT sample, ranging from 0.01 to 100 mg of freeze-dried rocket leaf powder/mL. Based on the UHPLC-MS/MS analysis, we used concentrations corresponding to a total polyphenol content of about 0.02–200 μM. The PAW-20 extract induced a significant proliferative effect compared with the control cells when used at a concentration of 50 mg/mL (100 μM), whereas, at a concentration of 100 mg/mL (200 μM), it caused a slight decrease in cell viability. In contrast, the extract of UT at the highest concentration tested resulted in a significant cytotoxic effect compared with the control. Comparing the effect of the UT and PAW-20 extracts, the greater cytotoxicity can be attributed to the former than the latter (see Figure 2).

This opposite effect on cell proliferation and cytotoxicity of extracts, derived from both *Diplotaxis tenuifolia* (L.) DC. and *E. sativa*, has been reported in the literature, albeit the assays were performed under different experimental conditions [36,37,38]. Interestingly, concentration-dependent activities were also reported in the literature for individual compounds such as erucin and sulforaphene (derived from the reaction of glucosinolates with the enzyme myrosinase) [39,40,41]. Furthermore, we cannot rule out a synergistic effect of the various bioactive compounds detected in the extracts.

### 3.4. Effect of PAW-Rocket Salad Extract on Cellular Redox Homeostasis

The results on cell viability prompted us to investigate the effect of the extracts on cellular redox homeostasis. In a healthy state, both reactive oxygen species (ROS) and reactive nitrogen species (RNS) are generated in a well-regulated manner, controlling cellular functions by modulating signaling pathways and playing an important role as second messengers. Oxidative stress is characterized by an imbalance between increased levels of ROS/RNS and low activity of cellular radical scavenging mechanisms. Oxidative stress contributes to aging and plays a role in the development of different human diseases (for example diabetes, cancer, and Alzheimer’s disease) [42,43]. 

The production of ROS was evaluated using the DCFDA Cellular ROS Detection Assay Kit (Abcam). Caco2 cells were incubated for 5 h in the presence of the extracts at the concentrations tested for the cell viability assays. We observed a significant increase in intracellular ROS production in a concentration-dependent manner for both the PAW-20 and UT extracts compared to the control cells (Figure 3a).

However, we highlighted a significantly lower ROS generation when the cells were treated with the PAW-20 compared to the UT extract, starting at a concentration of 5 mg/mL (10 μM) (Figure 3a). Interestingly, the highlighted ROS production only partially affected cell viability. These results are in good agreement with our previous study, in which we reported that intracellular ROS production was lower in Caco2 cells receiving polyphenol extract derived from apples exposed to DBD plasma technology than in the untreated ones [25]. Then, we determined the modulation of the intracellular NO levels, following the cells’ treatment in the same experimental conditions reported above. NO was assessed by measuring the stable intracellular oxidation products nitrite and nitrate, using the Nitric Oxide Assay Kit (Abcam). Besides its role in triggering redox imbalance, NO is a ubiquitous mediator of many different biological processes, such as vasodilation, neurotransmission, and immune response [44]. Moreover, NO modulates intestinal epithelial cell tight junction and plays a role in gastrointestinal motility, both under physiological and pathological conditions [45]. As previously revealed for ROS production, we demonstrated an increased generation of NO in a concentration-dependent manner for both the PAW-20 and UT samples, compared to the control cells. Moreover, the intracellular NO production was, in any case, lower in the Caco2 cells loading with the PAW-20 than the UT extract (Figure 3b). 

The modulation of oxidative stress by *E. sativa* extracts has been poorly investigated, and sometimes the results seemed to be conflicting. Treatment of human peripheral blood mononuclear cells with an *E. sativa* extract or glucosinolate fraction does not induce a significant modulation of ROS production, while it is able to reduce the cytotoxicity and ROS production induced by H_2_O_2_ treatment [46]. Other authors also reported the antigenotoxic effect of both glucosinolate-rich extract of *E. sativa* cv. Sky against H_2_O_2_ [47] and *E. sativa* extract against benzo[a]pyrene-DNA-induced damage [48]. However, it is documented that bioactive compounds derived from different species of the Brassicaceae family have the ability to induce oxidative stress in cancer cells [49]. Indeed, the notion that ROS/RNS are ‘bad’ or ‘good’, in this context, needs to be further elucidated.

According to the obtained results, the use of the PAW technology led to an immediate slight increase in the RSA analyzed in the amphiphilic fraction compared to the UT and the CL samples, the latter being used as an industrial process reference. On the other hand, the PAW determined a higher value of the reducing power with increasing treatment time. It is worth mentioning that our results revealed a significantly greater relative abundance (*p* ≤ 0.05) of glucosinolates in the PAW-20 sample compared to the untreated one. Since the literature data suggest that these unique phytochemicals, and their related isothiocyanates, represent an important contribution to human health, further studies are needed to clarify the effects of PAW technology on the key enzymes involved in the glucosinolate pathway. Furthermore, given the key role of oxidative and nitrosative stress in the development and progression of various human diseases, these preliminary results should be pursued.

## 4. Conclusions

The results obtained in this study indicate that the immersion of rocket in PAW reduced very few phenolic compounds compared to the untreated sample (UT), while the use of PAW technology seemed to positively affect the relative percentage of glucosinolate. However, the opposite effects on cell viability observed in the PAW-20 and UT extracts could be explained by the differences observed in the polyphenol profile. The data obtained in human cultured colonocytes showed that the polyphenol extract obtained from rocket leaves did not induce a significant change in cell viability after exposure to the PAW, in contrast to the extract obtained from the UT, which, on the contrary, induced a cytotoxic effect at the highest concentration tested. On the other hand, both extracts induced an imbalance in the Caco2 cell redox status, albeit the PAW extract exhibited a lower effect.

In conclusion, the *in vitro* and in-the-cell-line results provide new insight into the effects of PAW technology on food matrices, for its potential application as a novel and safe strategy in the food industry.

## Figures and Tables

**Figure 1 nutrients-14-05337-f001:**
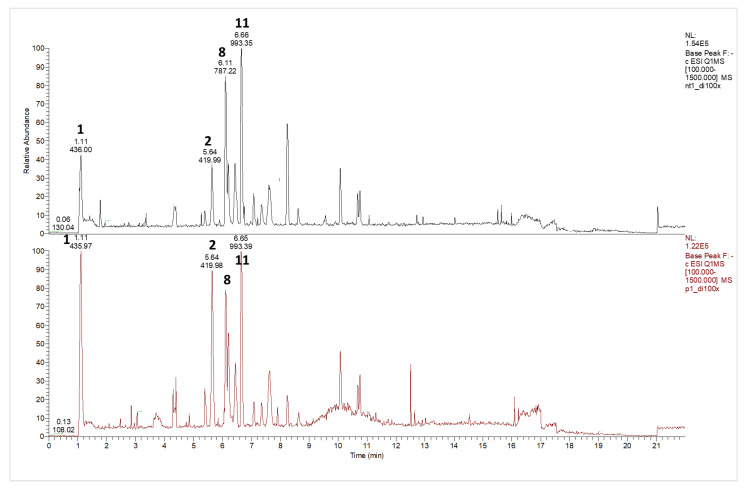
UHPLC-MS/MS full-scan chromatograms (RT: 0.00 – 21.99 min) of the methanolic extracts of rocket salad, washed in PAW for 20 min (PAW-20, in red) or untreated sample (UT, in black). Numbers indicate the most abundant peaks (as relative ion abundance), and their corresponding names are reported in Table 2. Chromatographic peak detection for [M-H]^−^ at m/z 436 (peak 1), 420 (peak 2), 787 (peak 8), and 993 (peak 11) are reported in Appendix A.

**Figure 2 nutrients-14-05337-f002:**
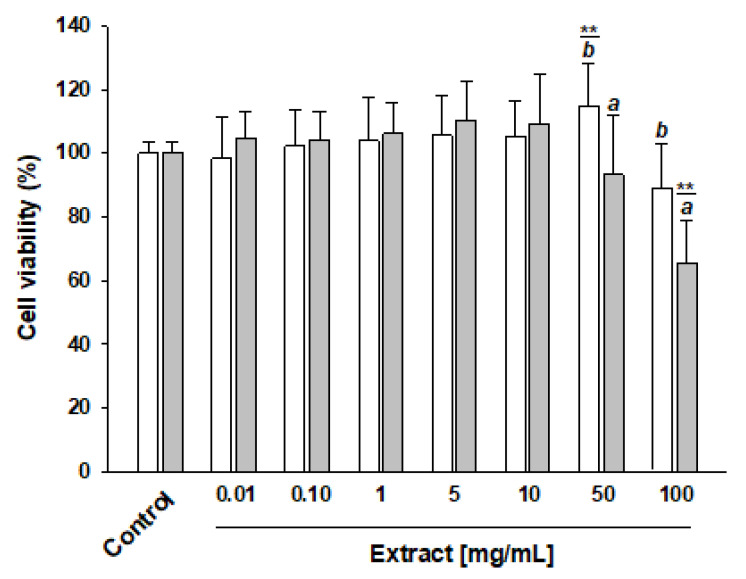
Effect of PAW-rocket salad extract on Caco2 cell viability. Caco2 cells were treated for 5 h with increasing concentrations of polyphenols (0.01–100 mg/mL) extracted from PAW-20 (white bar) or UT (gray bar) rocket salad. Cell viability was determined by CellTiter-Glo^®^ Luminescent Cell Viability Assay (Promega). Control represents Caco2 cells incubated with culture medium containing 0.1% DMSO. Data are presented as the mean ± SD of relative percentage of control sample (set at 100%) obtained from two independent experiments each carried out in quadruplicate. Statistical significance was calculated by pairwise multiple comparisons for one-way ANOVA followed by Tukey’s HSD post hoc test. **: significant difference versus control *p* < 0.001; different letters, when reported, mean significant difference between groups (PAW-20 versus UT at different concentrations), *p* < 0.05.

**Figure 3 nutrients-14-05337-f003:**
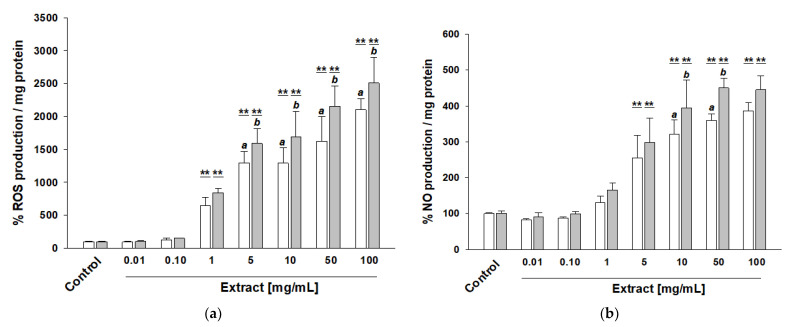
Effect of PAW-rocket salad extract on cellular redox homeostasis. Caco2 cells were treated for 5 h with increasing concentrations of polyphenols (0.01–100 mg/mL) extracted from PAW-20 (white bar) or UT (gray bar) rocket salad. (**a**) Intracellular ROS production was determined by DCFDA ROS Assay Kit (Abcam). (**b**) Intracellular NO production was determined by the Nitric Oxide Assay (Abcam). Control represents Caco2 cells incubated with culture medium containing 0.1% DMSO. Data are presented as the mean ± SD of relative percentage of control sample (set at 100%) obtained from two independent experiments, each carried out in quadruplicate. Statistical significance was calculated by pairwise multiple comparisons for one-way ANOVA followed by Tukey’s HSD post hoc test. **: significant difference versus control *p* < 0.01; different letters, when reported, mean significant difference between groups (PAW-20 versus UT at different concentrations), *p* < 0.05.

**Table 1 nutrients-14-05337-t001:** Antioxidant activity in rocket salad samples evaluated by different *in vitro* methods as affected by PAW washing for different times. Data are expressed on dry matter (DM) as mean ± standard deviation (SD) of triplicate analysis.

Sample	Time (min)	RSA_ABTS_ (µmol TE g^−1^ DM)				RSA_DPPH_ (µmol TE g^−1^ DM)		FRAP (µmol Fe^2+^ g^−1^ DM)		TPC (µmol GAE g^−1^ DM)			
		Hydrophilic Extract		Amphiphilic Extract		Amphiphilic Extract		Amphiphilic Extract		Hydrophilic Extract		Amphiphilic Extract	
		Mean	SD	Mean	SD	Mean	SD	Mean	SD	Mean	SD	Mean	SD
UT	-	44 ^b^	4	50 ^a,b,c^	6	29 ^a^	7	39 ^a,b,c^	6	22 ^b^	2	26 ^a^	5
CL	-	54 ^b^	4	42 ^a,b^	11	34.4 ^a,b,c^	0.8	53 ^c^	9	25 ^b^	1	62 ^c^	6
PAW	2	31 ^a^	3	65 ^c^	3	42 ^c^	1	32 ^a^	2	14 ^a^	1	34 ^a,b^	5
PAW	5	43 ^a,b^	4	44 ^a,b^	1	27 ^a^	2	36 ^a,b^	5	21 ^b^	2	29 ^a^	3
PAW	10	53 ^b^	7	56 ^b,c^	6	40 ^b,c^	2	45 ^a,b,c^	4	24 ^b^	2	32 ^a^	5
PAW	20	51 ^b^	4	39 ^a^	3	31 ^a,b^	3	47 ^b,c^	2	19 ^a,b^	3	46 ^b^	6

UT: untreated sample; CL: hypochlorite solution washing; PAW: sample washed in Plasma Activated Water; RSA: Radical Scavenging Activity; TE: Trolox Equivalents; FRAP: FeSO_4_ equivalents; TPC: Total Phenolic Content; GAE: Gallic Acid Equivalents. Data marked with the same letter within each column are not significantly different (Tukey’s HSD, *p* > 0.05).

**Table 2 nutrients-14-05337-t002:** Chromatographic characterization of compounds identified by UHPLC-MS/MS in the methanolic extracts of rocket salad sample washed in PAW for 20 min or untreated sample (UT).

Peak	Compound ^1^	RT (min)	[M-H]^−^ (m/z)	MS/MS Ions (m/z) ^2^	References
1	glucoraphanin	1.11	436	97, 96, 178, 194, 227, 259, 275	[29]
2	glucoerucin	5.64	420	97, 96, 75, 259, 275, 178, 227, 242,195	[29]
3	isorhamnetin-*o*-hexoside	7.15	477	314, 315, 285, 271, 299, 300, 243	[22]
4	kaempferol-*o*-dihexoside	6.45	609	446, 284, 285, 283, 483, 327	[21]
5	quercetin-*o*-dihexoside	6.40	625	463, 301, 300	[23]
6	isorhamnetin-*o*-dihexoside i	6.47	639	315, 313, 476, 477	[21]
7	isorhamnetin-*o*-dihexoside ii	7.03	639	315	[21]
8	quercetin-*o*-trihexoside	6.11	787	463, 625, 301	[21]
9	quercetin-*o*-dihexoside-*o*-feruloyl-*o*-hexoside i	6.74	963	801, 639, 463, 301, 625	[21]
10	quercetin-*o*-dihexoside-*o*-feruloyl-*o*-hexoside ii	7.03	963	801, 639, 463	[21]
11	quercetin-*o*-dihexoside-*o*-sinapoyl-*o*-hexoside	6.66	993	831, 626, 670, 463, 301, 669	[21]
12	kaempferol-*o*-hexoside	7.06	447	284, 285, 255, 257	[22]

RT: Retention Time. ^1^ Identification of compounds is accomplished with mentioned reference materials; ^2^ MS/MS ions are reported as a function of their relative abundance.

**Table 3 nutrients-14-05337-t003:** Polyphenol concentrations (µmol/L) determined in methanolic extracts of rocket salad washed in PAW for 20 min (PAW-20) or in the untreated sample (UT). Values are reported as means of three replicates ± standard deviation (SD).

Compounds	UT		PAW-20		Sig. ^1^
	Mean (µmol/L)	SD	Mean (µmol/L)	SD	
kaempferol-*o*-hexoside	n.q.	-	n.d.	-	-
isorhamnetin-*o*-hexoside	78	6	13	4	*
kaempferol-*o*-dihexoside	126	6	118	19	-
quercetin-*o*-dihexoside	101	13	73	3	*
isorhamnetin-dihexoside i	194	18	223	22	-
isorhamnetin-dihexoside ii	n.q.	-	n.q.	-	-
quercetin-*o*-trihexoside	845	69	749	54	-
quercetin-*o*-dihexoside-*o*-feruloyl-*o*-hexoside i	21	3	22	8	-
quercetin-*o*-dihexoside-*o*-feruloyl-*o*-hexoside ii	n.q.	-	n.q.	-	-
quercetin-*o*-dihexoside-*o*-sinapoyl-*o*-hexoside	833	105	838	154	-
total polyphenols	2198	219	2036	264	-

^1^*t*-test, * *p* ≤ 0.05; n.q.: lower than the LOQ (at 16.3 µmol/L); n.d.: lower than the LOD (at 0.2 µmol/L).

## Data Availability

Not applicable.

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
