# Peer review of "Fresh-Cut Eruca Sativa Treated with Plasma Activated Water (PAW): Evaluation of Antioxidant Capacity, Polyphenolic Profile and Redox Status in Caco2 Cells"

_nutrients, 2022, doi:10.3390/nu14245337_

Round 1

Reviewer 1 Report

TITLE: Fresh-cut Eruca sativa Treated with Plasma Activated Water  (PAW): evaluation of Antioxidant Capacity, Polyphenolic Pro-3 file and Redox Status in Caco2 cells

Comments to authors:

This study aimed to the effect of Plasma Activated Water (PAW) treatment on the Antioxidant Capacity, Polyphenolic Pro file and oxidative status in Caco2 cells of fresh-cut rocket salad. The work has some merits and within the scope of Nutrients, and valuable to the journal’s readership. However, the manuscript needs revisions and the suggestions are revealed as follows.

1. The introduction is too long, it should be condensed. Novelty of the present manuscript could be stated more clearly in the introduction section in the manuscript.

2. There are some similar section for the aurhors’ previous published article “Ramazzina, I., Berardinelli, A., Rizzi, F., Tappi, S., Ragni, L., Sacchetti, G., & Rocculi, P. (2015). Effect of cold plasma treatment on physico-chemical parameters and antioxidant activity of minimally processed kiwifruit. Postharvest Biology and Technology, 107, 55-65.” Please revise it.

3. Figure 1 is blurry, please revise it.

4. The concentration of extract treated on cell is too high (50, 100 mg/mL). The design and results of cell experiment is simple and not good. The phenolic extract treatment should provide protection for cell due to their antioxidant activity, however, the result shows that rocket salad extract promote the production of ROS and NO? The significant analysis should be performed between treatment groups and control.

5. There are some grammars errors in the manuscript, please check throughout the manuscript and to ensure the grammars are correct.

6. P should be revealed as small letter in the manuscript.

7. The glucosinolates levels of rocket salad should be expressed as μmol/g DM. The method of calibration curve was adopted for quantification of flavonol glycosides by using rutin (external standard) calibration, thus the result is relative content and the expression should be revised.

Author Response

Reviewer 1

Comments to authors:

This study aimed to the effect of Plasma Activated Water (PAW) treatment on the Antioxidant Capacity, Polyphenolic Pro file and oxidative status in Caco2 cells of fresh-cut rocket salad. The work has some merits and within the scope of Nutrients, and valuable to the journal’s readership. However, the manuscript needs revisions and the suggestions are revealed as follows.

  1. The introduction is too long, it should be condensed. Novelty of the present manuscript could be stated more clearly in the introduction section in the manuscript.

Introduction has been shortened.

  1. There are some similar section for the aurhors’ previous published article “Ramazzina, I., Berardinelli, A., Rizzi, F., Tappi, S., Ragni, L., Sacchetti, G., & Rocculi, P. (2015). Effect of cold plasma treatment on physico-chemical parameters and antioxidant activity of minimally processed kiwifruit. Postharvest Biology and Technology, 107, 55-65.” Please revise it.

The cited article has definitely some common features with the present one, since the methodological approach is very similar, although we are applying it to a different raw material but also using a different type of plasma treatment. However, in the revised version, we tried to modify as much as possible the description of the methods and the results.

  1. Figure 1 is blurry, please revise it.

Figure 1 has been replaced

  1. The concentration of extract treated on cell is too high (50, 100 mg/mL). The design and results of cell experiment is simple and not good. The phenolic extract treatment should provide protection for cell due to their antioxidant activity, however, the result shows that rocket salad extract promote the production of ROS and NO? The significant analysis should be performed between treatment groups and control.

We thank the Reviewer for this comment. Based on HPLC-MS/MS analysis, we used concentrations corresponding to a total polyphenol content of about 0.02-200 μM. We chose a broad range of the concentration to verify both the proliferative/cytotoxic effect and the oxidative stress induced by extracts. Indeed, literature data report that these effects are concentration-dependent (as reported in line 388-397). Specifically, the chemical reactivity of polyphenols makes these compounds susceptible to generate reactive oxygen species, exhibiting pro-oxidant effects. So, polyphenols can exert both a scavenging activity (usually at low concentration) and a pro-oxidant actions (usually under high-dose conditions) (Nowak et al., 2022 https://doi.org/10.3390/molecules27113453; Kanner et al., 2020 https://doi.org/10.3390/antiox9090797; León-González et al., 2015 https://doi.org/10.1016/j.bcp.2015.07.017).

With respect to statistical analysis, we firstly performed a comparison between control cells and treated cells, just to evaluate the effect of the polyphenolic extracts on ROS/RNS productions. Next, we compared PAW treated and untreated extracts, because the aim of our research was to evaluate the impact of PAW treatment on food matrix. In agreement, we improved “Statistical analysis” section at lines 232-234; the statistical analysis were also reported in the captions.

  1. There are some grammars errors in the manuscript, please check throughout the manuscript and to ensure the grammars are correct.

The English language has been revised throughout the manuscript

  1. P should be revealed as small letter in the manuscript.

P has been corrected.

  1. The glucosinolates levels of rocket salad should be expressed as μmol/g DM. The method of calibration curve was adopted for quantification of flavonol glycosides by using rutin (external standard) calibration, thus the result is relative content and the expression should be revised.

Authors thank the reviewer for this suggestion. However, the use of rutin (rutoside) calibration method was specific for the quantification of flavonol glycosides. In M&M section (at line 148 and 177), author specified that the HPLC-MS/MS quali-quantitative method was applied for polyphenolic compounds (specifically for flavonol glycosides) in the analysed extracts.

In addition, in these analytical conditions, glucosinolates were detected, and their relative percentages were performed for mainly comparison purposes between the extracts from PAW treated and untreated samples. Interestingly, next step will be to investigate how the PAW affected the further enzymatic hydrolysis of glucosinolates into their corresponded products.

Author revised throughout the text to underline that analytical conditions were specifically performed for the polyphenolic fraction analysis.

Reviewer 2 Report

In this manuscript authors discussed the  Fresh-cut Eruca sativa Treated with Plasma Activated Water 2 (PAW): evaluation of Antioxidant Capacity, Polyphenolic Pro- 3 file and Redox Status in Caco2 cells but following should be considered.

1. The abstract shoul have the important findings value should be given.

2. line 37-38  Many phenolic compounds are antioxidants that  may aid in the prevention of human diseases (e.g., cancer and heart diseases) and they also posses immunomodulatotry activity authors may go through https://doi.org/10.1016/j.sajb.2022.04.055.

3. References is needed for Total phenolic content and antioxidant activity.

4. Comprasion with the standard drug is needed. like ascobic acid, BHT.

5. Possible mechanism should be discussed.

6.  HPLC data may be given to support HPLC.

7. Indicate the major constituents in uhplc mass data.

8. If possible support the data using insilico or invivo approach.

9. Check for spelling and english errors.

10. Strictly follows english grammar.

Author Response

In this manuscript authors discussed the Fresh-cut Eruca sativa Treated with Plasma Activated Water 2 (PAW): evaluation of Antioxidant Capacity, Polyphenolic Pro- 3 file and Redox Status in Caco2 cells but following should be considered.

  1. The abstract shoul have the important findings value should be given.

The abstract has been revised adding same numerical values related to the results.

  1. line 37-38 Many phenolic compounds are antioxidants that may aid in the prevention of human diseases (e.g., cancer and heart diseases) and they also posses immunomodulatotry activity authors may go through https://doi.org/10.1016/j.sajb.2022.04.055.

We thank the Reviewer for this comment. In agreement, we improved the main text with the following sentence (Lines 41-44): “Many phenolic compounds are antioxidants that may aid in the prevention of human diseases (e.g., cancer and heart diseases), and they also possess immunomodulatory activity. Health promoting effects of a vegetable-rich diet have partly been attributed to increased intake of phenolic compounds with high antioxidant capacity (Jideani et al., 2021; Shukla et al., 2022)”.

  1. References is needed for Total phenolic content and antioxidant activity.

We thank the Reviewer for this comment. In agreement, we improved the main text in the section “2.4 Total phenolic content and antioxidant activity” (Line 126)

  1. Comprasion with the standard drug is needed. like ascobic acid, BHT.

Authors thank the reviewer for his/her suggestion. However, the objective of the present study was to evaluate how PAW treatment affected fresh-cut rocket salad health-promoting properties among these the antioxidant capacity, thus not the measurement of antioxidant activity in comparison with an antioxidant substance such as ascorbic acid or BHT. So, all the in vitro analyses were performed using the appropriate standard, as previously reported in Ramazzina et al., 2015.

.

  1. Possible mechanism should be discussed.

More discussion of the possible mechanism has been added in the revised version.

  1. HPLC data may be given to support HPLC.

Authors added HPLC data as Supplementary Material (see Supplementary file S1: chromatograms and data file).

  1. Indicate the major constituents in uhplc mass data.

Authors specified the major peaks [m/z 436 (peak 1), 420 (peak 2), 787 (peak 8) and 993 (peak 11)] detected in full scan chromatograms in Figure 1 caption and added extracted chromatograms for [M-H]-  at selected m/z in Supplementary File S1.

  1. If possible support the data using insilico or invivo approach.

We thank the Reviewer for this suggestion. In a previous study we evaluated the effect of PAW technology on different endogenous spoilage microorganisms, highlighting the potential use of PAW treatments in the food industry (Laurita et al., 20211). Thanks to the results obtained in this study, we demonstrated that in Caco2 cells low concentrations of the polyphenolic extract derived from PAW treatment did not induce neither a cytotoxic effects nor a redox status. Moreover, we evaluated a positive effect of PAW on glucosinolate content. Overall, the next step of our research is to move towards an in vivo approach to confirm the safety, the qualitative and nutritional characteristics of food matrices treated with PAW.

  1. Check for spelling and english errors.
  2. Strictly follows english grammar.

The English language has been revised throughout the manuscript

Round 2

Reviewer 1 Report

The authors well revised the manuscript. 

Reviewer 2 Report

Comments incorporated